# The experience of the COVID-19 pandemic by persons with ASD: Social aspects

Jacek Błeszyński[1¤a‡], Aleksandra Rumińska[2☯*], Agnieszka Hamerlińska[3‡], Renata Stefańska-Klar[4‡], Agnieszka Warszawa[5☯¤b]

1 Faculty of Philosophy and Social Sciences Institute of Pedagogical Science, Nicolaus Copernicus University, Toruń, Poland, 2 Doctoral School, University of Silesia, Katowice, Poland, 3 Faculty of Philosophy and Social Sciences Institute of Pedagogical Science, Nicolaus Copernicus University, Toruń, Poland, 4 Institute of Educational Studies, State Higher Vocational School, Racibórz, Poland, 5 Autism Team Foundation, (Institutum Investigationis Scovorodianum), Łódź, Poland

☯ These authors contributed equally to this work.
¤a Current address: Faculty of Pedagogical Sciences, Cardinal Stefan Wyszynski University, Warsaw, Poland
¤b Current address: Institutum Investigationis Scovorodianum (in 2020 as part of the Autism Team Foundation), The Gregory Skovoroda Institute for Scientific Research of Autism, Łódź, Poland
‡ JB, AH and RSK also contributed equally to this work.
* aleksandra.ruminska7802@gmail.com

**Data Availability Statement:** All relevant data are within the manuscript and its Supporting Information files.

## Abstract

While causing a variety of social restrictions, the COVID-19 pandemic has also precipitated the digitalisation of public services and official procedures, reducing many, until recently necessary, immediate social interactions. This study has been conducted to investigate their perception of the COVID-19 pandemic and its impact on their current and future social interactions. To this end, semi-structured narrative interviews were conducted. Ten adults on the autism spectrum participated in the study. The phenomenological analysis of the narratives focused on categories related to the social functioning of the study participants. The interpretation of the narratives has shown that autistic people can experience a sense of loss due to the lack of direct contact. On the other hand, we also talked to the participants who expressed their satisfaction with the situation of obligatory social distance. The respondents also discussed the subject of changing the form of interaction in some areas of public life to one that is more adjusted to the needs of people with their condition. The study concludes with a suggestion that autistic people might benefit from technological progress in institutions and the availability of the option to prefer online contact for interactions that are not strictly necessary.

## Introduction

On 17 November 2019, the SARS-CoV-2/COVID-19 epidemic first broke out in Wuhan, a city in the province of Hubei in central China. The information about the novel virus spread immediately in the media, and on 11 March 2020, the World Health Organisation (WHO) declared the pandemic. Countries across the globe launched actions to fight the threat and

**Funding:** During the peer-review process, the CRUSH project, which was our main source of funding, was already in need of settlement. The current situation is as follows: Funds from University of Silesia from Poland: 625 USD

**Competing interests:** AUTHORSHIP STATEMENT Manuscript title: The Experience of the COVID-19 Pandemic by Autistic People: Social Aspects All persons who meet authorship criteria are listed as authors, and all authors certify that they have participated sufficiently in the work to take public responsibility for the content, including participation in the concept, design, analysis, writing, or revision of the manuscript. Furthermore, each author certifies that this material or similar material has not been and will not be submitted to or published in any other publication before its appearance in the Autism. Authorship contributions: Category 1 Conception and design of study: Aleksandra Rumińska, Agnieszka Warszawa; acquisition of data: Aleksandra Rumińska; analysis and/or interpretation of data: Aleksandra Rumińska, Agnieszka Warszawa. Category 2 Drafting the manuscript: Aleksandra Rumińska, Agnieszka Warszawa; revising the manuscript critically for important intellectual content: Jacek J. Błeszyński, Aleksandra Rumińska, Agnieszka Hamerlińska, Renata StefańskaKlar, Agnieszka Warszawa. Category 3 The Research Project Manager: Jacek J. Błeszyński; Approval of the version of the manuscript to be published (the names of all authors must be listed): Jacek J. Błeszyński, Aleksandra Rumińska, Agnieszka Hamerlińska, Renata Stefańska-Klar, Agnieszka Warszawa. Acknowledgements All persons who have made substantial contributions to the work reported in the manuscript (e.g., technical help, writing and editing assistance, general support), but who do not meet the criteria for authorship, are named in the Acknowledgements and have given us their written permission to be named. If we have not included an Acknowledgements, then that indicates that we have not received substantial contributions from non-authors: Wojciech Kruszelnicki, Ewa Jarosz.

implemented preventive measures. Restrictions were introduced that have significantly changed the functioning of societies. In March 2020, the Polish Ministry of Health announced recommendations to protect people from the spread of the virus. Social contacts, both those held for pleasure and those related to health, official matters, shopping, or work, have largely moved to the Internet or have been limited to phone calls and emails.

While the pandemic affects entire societies, our study focuses on the population of people on the autism spectrum, the diagnostic criteria for which include persistent deficits in social communication and social interaction [1–3]. Steven K. Kapp [4] argues that the social needs of people on the spectrum are quite diverse and depend both on biological factors, such as sensory sensitivities and environmental factors, which can shape the need for control or motivation. While we agree with Kapp's position, we decided to investigate how this particular group is dealing with the current pandemic restrictions.

Research by Kristen Gillespie-Lynch [5] indicates that online interaction contributes to increased comprehension and control over communication in autistic people. It is also said to reduce the emotional, social and time pressures experienced by them in offline situations [6, 7]. People on the autism spectrum function in communities of non-autistic people, most of whom do not experience any difficulties related to social functioning. A study of British adults found 1% of respondents to be autistic [8]; however, the number of diagnosed adults continues to grow every year, and the population of women on the spectrum seems to be underestimated [9]. Given this data and specific social needs of autistic people, we decided to study their perception of the changed form of contact and expectations regarding institutional solutions dedicated to people on the spectrum.

## Methodology

### Researchers

The study and data analysis were conducted by an interdisciplinary team of employees from the Nicolaus Copernicus University in Toruń, the University of Silesia, and the *Institutum Investigationis Scovorodianum* (in 2020 as part of the Autism Team Foundation) in Łódź as part of a project the CRUSH (Research on the COVID-19 pandemic, Project No. 2/2020/FT). Members of the team included autistic as well as non-autistic researchers.

The aim of the study was to establish how people on the autism spectrum perceive the impact of the COVID-19 pandemic on their social interactions, both present and future (after the pandemic), by analysing the respondents' narratives. To this end, the study focused on recognising the meanings assigned to certain elements of reality in respondents' interpretations of that reality. Our research task was to understand the sense hidden in the participants' attempts to make their experiences meaningful. The double hermeneutic was adopted as the general theoretical scaffold [10], with interpretative phenomenological analysis (IPA) laying the foundations for the methodology. No hypotheses were formulated; neither was the study launched with a prior theory in mind or any pre-existing data for analysis. Initially, only research problems were identified. It was only upon the analysis of the interviews that researchers sought to make out how respondents experienced the changes caused by the COVID-19 pandemic.

The following research problems were identified:

a.  What experiences related to the COVID-19 pandemic are reported by study participants? How significant are they in terms of respondents' social functioning? Which aspects of autistic people's lives have improved during this time?

b.  What problems related to social functioning are reported by respondents?

c. Which solutions, implemented during the pandemic, could improve the quality of life of the respondents in the post-pandemic reality?

While conducted for research purposes, the study also had a practical angle. It intended to identify specific solutions in the field of social communications, such as digitalisation, which may be implemented on a regular basis to improve the functioning of autistic people in society.

Prior to its implementation, the project was approved by the Research Ethics Committee of the Nicolaus Copernicus University in Toruń (application 2/2020/FT, decision of 14 June 2020).

## Study participants and environment

Non-probabilistic sampling was used for the purposes of qualitative research. Participants were selected by way of stratified random sampling to obtain the most heterogeneous sample. Prior to the study, all participants expressed their oral consent, and later they confirmed with a signed consent form. Ten adults diagnosed with autism spectrum, aged 25–45, were invited to take part in the study, out of whom six identified themselves as female, two as male, and two as non-binary—the grammatical forms in the text below correspond to those used by participants [11, 12].

When planning the study, we predicted that certain factors needed to be considered upon the selection of participants to ensure the sample variety and therefore diverse perceptions of the pandemic experiences. In order to provide a better understanding of the sample's profile, this section describes the respondents' family, housing, work, and financial situation.

All participants are within the intellectual norm or have supra-normative cognitive abilities. Like many people on the autism spectrum, they may have anxiety disorders or depression [13–15].

They were qualified on the basis of self-report, but their intellectual status was confirmed by the organizations that made contacts with each of them available.

Three participants started their own families, one had a partner, six were single, with three of them sharing a flat with their parents. Six respondents had higher education (two of them had bachelor degrees), four had secondary education, two were still continuing their education. Eight participants were professionally active, one worked as a volunteer, and one was unemployed. Six participants described their income as below PLN 2,500 net, with three of them partially relying on the financial support from parents, while four respondents reported a net income of PLN 3001–4000. All participants lived in Poland: three in towns (population of 20,000–100,000) and seven in cities (population of 100,000 or more).

Semi-structured narrative interviews (with the use of several previously prepared questions) were conducted. This research method requires a relationship between the researcher and participants, who share their views of the reality as experienced by them, including any biographical and social information that is important to them. Given that participants were autistic, the interviews were also conducted by a person on the spectrum who was attentive to respondents' need to reflect longer on a question or have more details to ensure that statements were as complete as possible and given freely. With a high likelihood of sensory sensitivities, high-quality equipment was ensured and background distractors were minimised.

Participants were asked the following questions:

1. How has your social life changed due to the COVID-19 pandemic in terms of your contacts at work and other social interactions?

2. Has your condition changed throughout the pandemic? How? Which emotions would you identify as dominant?

3. What do your social contacts look like during the pandemic? How well do you manage with initiating and maintaining social contacts?

4. What strategies help you cope with the present situation?

5. Can you see any positive sides of what the current reality looks like?

6. What difficulties related to the current situation can you identify?

7. What legal or institutional support and solutions would help you deal with these difficulties?

8. How do you see your future after the pandemic?

The results obtained from the analysis of the narrative interviews are presented below.

## Procedure

Individual interviews were conducted between 22 October and 13 November 2020 through an online communication platform provided by the University of Silesia in Katowice.

Each interview started by establishing rapport with the respondent. The participants were given unlimited time, and attention was paid to the conversation space. The interviews were scheduled within a short period to reflect one stage of the pandemic–at the time when it was expected that the Polish government would implement a second lockdown [16] due to the increasing numbers of infections. In this way, all respondents could refer to similar experiences related to the COVID-19 pandemic. The interviews were recorded and stored as audio-visual files that were later transcribed.

## Data analysis

The collected data were subjected to an analysis based on the grounded theory methodology [17]. The following procedure was adopted: (a) transcription of interviews; (b) preparation of thematic codes; (c) selective coding of transcription by social aspects; (d) data grouping. Interviews were transcribed phonetically whereby their original form and potential implications for data coding and intonation analysis were preserved with the use of the Jefferson [18] Transcription System, S1 Table. All identifying information was removed and the participants' statements were coded as I1 to I10.

**The collected data were then subjected to a content analysis.** As a result, the following thematic categories were identified based on the recurring topics in the interviews:

1. Changes in the experience of social interactions in public space during the pandemic (education, work, medical care, culture, offices)

2. The form and intensity of family and social relations in the epidemic reality

3. Strategies for initiating and maintaining social relations in the epidemic reality

4. Perception of social behaviours in the context of the COVID-19 pandemic

5. Anticipated changes as a result of the pandemic experience

The data analysis allowed us to distinguish concepts that participants found particularly significant. Based on this, a theoretical model was developed.

### Community involvement statement

Out of five researchers participating in this project, two persons are autistic and involved in autistic communities. They conceived and designed this study, acquired the data, analysed and interpreted it, they also drafted the manuscript. The two researchers on the spectrum were involved in revising the manuscript critically for important intellectual content.

## Results

The participants approached the study with openness, which allowed us to obtain a significant amount of material with a variety of insights, both introspective and extrospective.

### 1. Changes in the experience of social interactions in public space during the pandemic (education, work, medical care, culture, offices)

In terms of the changes affecting their social interactions, the respondents talked about the pandemic restrictions connected with physical distancing and the need to adapt to preventive measures. Based on the participants' answers, it may be concluded that pandemic-related changes had an impact on the quality of their social interactions in various areas of life, such as education, work, and health.

**(I2)** Yyy . . . I did somehow switch to remote mode for example with learning English/ so for example I didn't give up learning English during the pandemic/ ashsh . . . (((snorts))) I'm not giving up, because I am still learning and so it goes. So yeah, in this respect . . . well, some things could just move to the Internet, but for example I couldn't move my yoga classes to the Internet/ (2) which are very important to me and which are . . . very important also for my health, for my, for my body, yeah [. . .].

Participants I1 and I4 described their negative experiences with online therapy and online education, difficulties with medical consultations, and even a loss of contact with a physician.

Some participants found a positive meaning in their experiences as regards their sensory sensitivities, special needs, and their ability to satisfy social needs, so characteristic of many people on the autism spectrum. As one of the respondents (I6) observed, moving certain activities to the Internet improved their accessibility for people with limited social needs, helping them to feel more accepted.

**(I6)** I hate going to shops .hh ((laughs)) so I don't miss that at all, yyy now I can buy () everything online iii yeah/ my husband won't yyy . . . he always said that I buy everything online, even if the shipping is more expensive, as long as I don't go anywhere, so now I can do so ((laughs)) without risking stupid comments, because this is safe. You don't have to go anywhere/ this is just fine with me that you don't have to go to to to the shops, because I hate it.

The author of the I5 narrative shared how she interpreted the rather vague instructions about social *distancing* literally [19], initially withdrawing entirely from social life, which resulted in her losing control over her life. The participant emphasised the importance of precise communication. She went on to argue that the instructions should recommend *physical distance and* caring for *social solidarity [emphasis mine—A.R.]*. Once she realised it, the respondent regained control and started taking action to improve her functioning in professional and social contexts in the new reality.

### 2. The form and intensity of family and social relations in the epidemic reality

The restrictions aimed at preventing the spread of the virus have had a significant impact on the social sphere, including the shape and the intensity of the respondents' social relations.

**(I1)** Oh yeah . . . well . . . practically speaking . . . with what is happening now, I'm already . . . yyy I'm having such thoughts really/ very unpleasant yyy . . . this is like . . . ((sighs)) like I regret for example that . . . I don't have some/ a specific dangerous tool at home, that . . . yyy that practically speaking it would be better to end it all and . . . / ((((eyes tear up))) and be done with it. (3) Ooor . . . these thoughts that . . . when I wanted to end it all, that I should have done it and/ not have those/ hopes for the future (2) but I'm also trying to put it all together in my own way. [. . .] Honestly, I. . . yyy. . . when it comes to needs, I had to go into the denial mode again. . . because if. . . yyy I thought about it. . . then. . . I would just fall apart. Yyy but . . . yyy well, I'm trying to keep/yy in touch with as many people as possible, yeah, on the web, just to write, yyy chat by yyy . . . because I'm still not convinced about mobile phones, mobile phones are evil ((smiles)), but . . . yyy just to chat on yyy . . . on Messenger yyy . . . or some other Zoom . . . yyy, this is well . . . well/ just it/ it it is important.

This excerpt conveys both suffering that led to the participant's suicidal thoughts and his strategies to regain control over his emotional and social life. He even started to initiate contacts, which until recently had been extremely difficult for him.

I2 and I5 reported elaborate social needs that could not be satisfied during the lockdown. I2 also talked about the sense of falling out of social roles.

I3 observed that with the pandemic restrictions in place, even the protests against the decision of the Polish Constitutional Tribunal (CT) criminalizing abortion provided an opportunity 'to do something with other people'.

**(I3)** [. . .] Meetings are cancelled, but yyy . . . when I . . . see people, it's like . . . they are meetings, they seem like meetings with friends, but they are somehow/ formalised, it's either . . . through the fantasy club or the team . . . [the Quidditch team] which is after all a formal thing. Well, I . . . for such meetings . . . or even games of the 'mahjong' club (2) which now can't take place. [. . .] Well, I . . . Well, I usually didn't go to all . . . the events I planned to attend (2) perhaps only half of what I . . . so to speak ((smiles)) should have attended . . ., well, and now it's all . . . everything I'd like to go to now is cancelled. So there you go . . .even that protest [against the CT decision about abortion restriction], regardless of the idea, I liked the fact that . . . I could get to do something with people ((laughs)) (4) Well, the studies, yeah . . .? Special education . . ., it is about getting to know people [. . .].

Another participant repeatedly emphasised the importance of family and friends in her life.

**(I5)** And then we got those mixed signals in May about the election and what have you, because summer was coming . . . and in June they said: 'Go on, live your lives!' . . . and they told everyone: 'Go vote and enjoy your holiday because . . . the virus is gone' (2) ((((in disbelief))) and I just sit there and think: 'What the f***? whaaat?!' (2) Well, excuse my language but I can't . . . these are my emotions and direct reflections, so . . . forgive me when I use swear words . . . # but seriously . . . # I sat there for like three weeks . . . in this state of (((with indignation and disbelief))): 'What the f***? what?! (2) What the f*** is going on here?! Really . . . Here they are, telling us to stay at home, to be careful, that I shouldn't go to my grandma, especially if I use, if I use . . . public transport, because my grandma is in a group of people at higher risk yyy . . . but I can go to work. 'So I go to work, but I can't go visit grandma, what the f***?!' It should be the other way round (((with indignation))), we should take care of these people! [. . .] No/ no. I haven't got over that yet . . . and I don't think I will . . . Yyy . . . ((sighs)) It made me sad now . . . ((wipes tears)) I miss my grandma very much, I miss her terribly, and I would love to just sit with her, goof around, talk nonsense, make dumplings and, I don't know, bake a cake, whatever . . . ((wipes tears)) Have some tea together . . . [. . .] because I don't know how much longer she will be with us. As I said, she is an elderly lady, with some health problems (3) And on the other hand, if I do go and meet her, and then three weeks later, touch wood, something happens and she dies, then . . . I'd never forgive myself and for the rest of my life I'd

know that I was the one who infected her with covid . . . killed her. Even if later they'd tell me that the tests were negative and showed nothing. I won't believe it anyway, because I am on the spectrum. I will continue to have my convictions . . . I know it . . . I know it is/ would be a heavy burden/ and I know that it is/ some of it is very irrational, but some of it is very rational . . . so the dissonance of this whole situation is excruciatingly painful because I really don't know what to do.

The author of this excerpt continued returning to the description of emotions related to the need to maintain physical distance. She emphasised her inability to understand it. Her objection, irritation, and suffering increased because of the need to, on the one hand, engage in social contacts related to professional life, but on the other, distance herself from a person he loved.

Another participant (I7) described her experience of sharing a living space with her husband, also on the spectrum, during the lockdown. Despite limited social needs, she tried to stay in touch with her friend through online platforms; however, this form of 'intermediated' contact proved to be unsatisfactory. The respondent also talked about the needs and limitations she had never realised before.

**(I7)** [. . .] .hh and we also have/ we need a lot of space iii my husband has always had this ritual of going to work and this . . . and he has no friends .hh because, you know, he's on the spectrum, I mean we kind of know-don't know, in this sense that he yyy . . . he has small social needs, and work was enough for him, but then it's like he doesn't have those friends from work, because/ my husband only talks about work () this is his special interest and it's very hard . . . ((laughs)) .hh to get him to talk about anything else . . . yyy . . . .hh he can't talk to me about programming, [. . .] Yyy . . . when it comes to my friends . . . Well, I don't have big social needs, but I do have regular friends . . . .hh and usually with my . . . friend, who is also on the spectrum, she's been diagnosed–this is how we met actually . . . .hh well, we would meet once a month and it . . . was enough to satisfy our social needs and, you know, now also yyy, we don't do it . . . we tried to meet online, but it didn't work, I guess it didn't satisfy . . . those needs of ours, .hh and . . . that friend of mine . . . has a lot of anxieties, so she's got a lot to think about . . . and, you know, this is how it is . . . [. . .] Yyy . . . come to think of it, I didn't even think about it before, but I actually miss going out once a month to. . . to meet someone [. . .].

At the beginning of her narrative, I9 mentioned strategies that helped her function in this space; however, eventually she did not specify her needs in terms of maintaining direct contacts with friends.

The next participant (I8) discussed her ambivalent attitude towards meeting people during the pandemic. On the one hand, she craved for some interaction, but on the other hand, direct contact with people during the pandemic made her anxious.

**(I8)** „ [. . .] I have contracted/ something like a social phobia, but . . . not a phobia of contact with people, but for example when yyy, I walk somewhere . . . even, even if I'm wearing a mask, but there is this crowd of people. Or when I'm not wearing a mask . . . .hh, you know, yy. . . like in the forest yy . . . and I can see someone approaching, it doesn't matter if they have a mask or not, then I just . . . I turn away, I run away yy . . . I hide somewhere behind the bushes . . . yyy . . . or cover my mouth with a handkerchief, my heart . . . is pounding, so that only this . . . this germ doesn't get to me. Or, for example, in the park . . . when I didn't put the mask on fast enough yyy . . . or I don't know, maybe I even had it on, but I would hold/ hold my breath when passing people . . . And that's so . . . on the one hand it is yy . . . justified because I'm afraid of the virus, on the other hand it's so . . . .hh (((embarrassed))) it's more like a panic attack and I know that yyy . . . it's not that it's so dangerous, if I actually passed someone without wearing a mask yy . . . but it gives me y, yy . . . the feeling of . . . being in control of the situation when I hold my breath. [. . .] The most annoying thing for me was that I just. . .

on the one hand, I want something, on the other. . . I'm scared when it came/ comes to meeting people. Whenever someone offered .hh yyy like a to go somewhere, but maybe I could . . . meet someone, I even had such . . . plans a couple of times, but now I'm putting all meetings with people off . . . .hh until after the pandemic.

Some participants have found the current situation neutral or even beneficial for their limited social needs. When talking about social restrictions, they pointed to their, also limited, professional contacts as fully satisfying their needs.

**(I4)** Well, yyy . . . you know . . . I don't I don't I don't have that that I must yyy like go out, you know? Yyy . . . Someone who doesn't know me very well can think that I am a very social person because I work with people and so to speak . . . I say that I work on people, eee, but my passion is psychological mechanisms, you know? Eee, I think you understand what I'm talking about and why I work with people ((laughs)). But in general most people make me furious aaa-and I don't feel the need to be with people [. . .].

A similar opinion was expressed by participant I6.

### 3. Strategies for initiating and maintaining social relations in the epidemic reality

Respondents also mentioned strategies of deliberately trying to move certain activities to a space excluded from the lockdown. Their narratives contained descriptions of actions used as substitutes for the suffered loss, a way to do something different or relieve stress during the lockdown. Nevertheless, our analysis focused specifically on forms of social relations. One participant (I2) explained that 'intermediated' social contacts did not meet his needs or expectations. In his opinion, they did not allow for the 'naturalness' of the relations which the author of the narrative cherishes.

**(I2)** [. . .] (. . .) for me . . . yyy . . . for me, I don't know . . ., a story like that when I sit at home in the evenings and watch Netflix and . . . and I . . . listen to music, well, this is no solution for me . . . [. . .] First, it's more difficult to have fun like this because (2) Zoom is not for, for partying. There is . . . those meetings also took place so rarely . . . also, yy also yyy they were . . . very tiring for people, so that this/ lack of physical contact and, for example, no possibility to gather together and hang out as a group, which is not possible on Zoom, yyy . . . this lack of natural . . . yeah . . . being natural in a relationship, it is, it is for them, well . . . It helps because you can talk to someone, but on the other hand, yy . . . this lack of natural contact takes the energy away. . .

Some participants had a positive experience with intermediated contacts, which is probably related to the fact that they did not have to worry about communication difficulties and maintaining control in social interactions.

**(I3)** [. . .] I mean . . . I rely even more on the Internet when it comes to contacts.

Due to this new, surprising, and unpredictable situation affecting many aspects of life, participants identified many strategies whose main objective was to try to regain control over life and mental balance.

### 4. Perception of social behaviours in the context of the COVID-19 pandemic

When asked what other experiences related to the COVID-19 pandemic, apart from the changes related to political strategies and information chaos, proved difficult for them, the participants pointed to the unpredictability of social behaviours. The authors of the narratives were often unable to separate the difficulties caused by the pandemic and the pandemic restrictions from those that were the result of dynamic changes in the socio-political space.

One respondent (I5) observed that when protective masks became obligatory, she did not stand out so much anymore, while social distancing and people's general greater care for hygiene allowed her to function better in public places and public transport [20].

**(I5)** I'm one of those people who appreciate it, yyy kind of yyy, and this physical distancing iii . . . taking care of yourself, your hygiene, I didn't think it was weird when Asians were wearing masks in the underground (2) for me it was strange that Poles didn't do that, and when I saw more people of other cultures wearing masks on the underground, I also started wearing a mask, even before it became trendy, so to speak (((ironic))) . . . You know . . . I wore a mask before when there was smog or on those days with yyy . . . of changing weather, when you could easily catch something, or for example–when I was under slept because I hadn't slept for two days, because my (((gestures energetically))) creative . . . juices just poured out of my ears and I, I don't know, had been sitting and painting something for/ two days, you know? Well, because you know . . . I have those spells, right? ((laughs)) Well, it's like . . . just put the mask on and it's all great, and people pay less attention to me. . . than before, because half of my earrings can't be seen, I am wrapped up, I have this part covered here, I wear black nitrile gloves, so you can't see the tattoos on my hands, so people don't stare at me that intensely. Nobody rubs against me on the underground, nobody shoves me or pushes me because it's not packed, because there can't be any crowds, yyy . . . [. . .]

Another participant (I10) had similar reflections.

## 5. Anticipated changes as a result of the pandemic experience

One participant (I9) expressed her awareness of the situation of some sectors in Poland as a consequence of the inevitable changes and overloaded systems due to the pandemic. She expressed her hope that this situation would enforce changes in priorities in the socio-political space, while indirectly improving the quality of her life. Two more participants (I3 and I5) shared her views. The fragment below clarifies this position:

**(I9)** [. . .] It highlighted the problems in . . . the health sector, in education in Poland . . . and thanks to this, I hope that . . . in a few years, maybe (2) the situation will be better in this country where I live because. . . this pandemic has shown people the real. . . yyy. . . deficits that have been present for years. Whether staff shortages, or some. . . logistic shortcomings iii (2) it will also make people realise that some things can really be done remotely . . . and it doesn't have to mean that online is of poorer quality, for example, education or . . . work. You don't have to sit in one place for hours, because if you don't then . . . you don't work hard enough or you are a demanding Millennial who doesn't know what they want . . . and maybe really the world . . . yy that has been created/ by Western society . . . it isn't. . . oh oh. . . (2) it has been built on something that can be easily challenged. . . for example by something as random as a pandemic. And suddenly it turns out that this yy . . . wonderful world . . . is falling apart. And . . . people too . . . yy how should I put it . . . Well, a positive thing . . . let's say so because I don't know if it counts, but for example, changes in people's awareness are positive because what I mean is that . . . it is positive if . . . I can live in a society that is more aware . . . it also can have positive outcomes for my health [. . .].

It was surprising to learn that in the face of the pandemic restrictions, the participants have noticed that non-autistic people who previously had no problem adapting to the abnormality of the new situation, in the course of time have found maintaining social contacts equally as challenging as it is for individuals on the spectrum. The authors of the narratives I5, I8, and I9 also pointed out that increasingly more non-autistic people sought help from specialists, while diagnosis, therapy, and pharmacological treatment were no longer a taboo but were becoming a common experience.

Researchers were also interested in the respondents' plans for the post-pandemic future. Their narratives about goals and intentions, along with emotions related to their plans ranging from excitement to scepticism, are a telling testimony to the respondents' currently unsatisfied needs, longings, and attitudes.

**(I1)** I Guess . . . ((smiles)) . . . I think that he yyy . . . yyy I would really yyy like to get our group . . . yyy together in some . . . place where you can really just sit there, have a chat yyy . . . just hug everyone, yyy get some yyy pizza, a bucket of wine or whatever you like, and just, you know, hang out together and talk ((smiles)).

However, not all participants shared the belief that the pre-pandemic reality would be back. Some of them were highly sceptical about it.

**(I10)** I don't have any plans . . . for 'after' because I don't think there is going to be any 'after'. I mean it would be nice/ and all. But. . . I am sceptical about people. Well. . . It can't be otherwise after what I have seen, I meet these people at work every day, so. . . I don't make plans for 'after'. But . . . I do have plans/ for the pandemic ((smiles)) Because . . . as I said, I'm studying all the time . . . Especially now ((laughs)) you can see how hard I'm studying now (((sarcastically))) Hm . . . (3) But when I'm not at work, I spend most of this time . . . studying and . . . soon I will be applying for a job in a completely different sector. Where I won't have to . . . um . . . have contact . . . with random people/ at work.

The most common wishes concerned re-establishing direct social contacts and returning to the pre-pandemic normality in this area; however, respondents also shared their doubts as to whether the pre-pandemic conditions as society would ever be back.

## Discussion

### Strengths

Our study had several strengths that contributed to the effectiveness of the research procedure and the resulting analysis. In qualitative research, a sample including 10 respondents is large enough as the analysis of narratives focuses on individual stories as evidence. Our respondents could speak freely and elaborate on every topic. As researchers, we did not suppress their stories, which allowed us to collect immensely valuable material. We listened to all narratives with due attention and empathy. We never got the impression that respondents became bored with the study or impatient. After The interviews were finished, our team listened to the narratives many times so as not to overlook any important details. As a result, in the spirit of cooperation and for the sake of credibility, we have generated significant amounts of data.

### Limitations

The greatest limitation of the study was our inability to conduct interviews in direct social contexts. Unfortunately, the formula of our study was affected by the epidemic situation and the restrictions. Nevertheless, it is our belief that the adopted formula proved beneficial for most participants, allowing them to maintain control and reduce the emotional, social, and time pressures otherwise experienced by them in direct interactions other than those dictated by individual social needs [6, 7].

### Practical implications

It is our hope that this study will inspire a less negatively stereotypical approach to social disability such as autism. Many autistic people seem to be well prepared for functioning in the new, rapidly emerging digital reality. However, this does not apply to all spheres of life or all people on the spectrum to the same degree. We hope that soon we will be able to return to

spontaneous, direct contacts, while considering the needs of us all as beneficiaries of digitalisation by ensuring further technological modernisation of workplaces and institutions which are currently operating in this mode effectively and without detriment to customers/citizens. Pellicano and Stears [21] have written about a revolution in digital services which is likely to outlive the pandemic itself.

## Recommendations for future research

This article is a continuation of the research on the functioning of people on the autism spectrum during the COVID-19 pandemic conducted as part of the CRUSH project. The first article prepared by our team, *Perceived stress and coping strategies of autistic people during the Covid-19 pandemic*: *A study report* (2021), is now in printing.

As our research has progressed, we have discussed the possibility to continue the exploration of the studied subject to further complement our findings. It is our intention to repeat the study in a new—post-pandemic—setting to verify to what extent the pandemic has changed the perception of the forms of interaction preferred by autistic people and the progress of digitisation in public spaces where direct interaction was previously unavoidable.

## Conclusion

It seems unjustified to perceive development in the autism spectrum as unequivocally associated with a social disability as demonstrated by the narratives shared with us by autistic people. The observed diverse social needs represented by respondents seem to confirm Steven K. Kapp's conclusions [4] regarding a high diversity of social needs and abilities among people on the spectrum. Support in this area requires an individual approach (getting to know, learn, and understand the needs of a specific person) rather than a universal one. The needs and capabilities of people on the autism spectrum may differ from the needs and capabilities of those who constitute the majority and as such have an effect on the functioning and health of people on the autism spectrum.

There are areas where it is possible to facilitate the functioning of people who need to have their social interactions limited to the strictly necessary ones, without detriment to society. In addition, such changes could provide these individuals with the feeling of more independence and autonomy, also after the pandemic.

The time of the pandemic has made us realise that official contacts, shopping, and even work through the Internet are possible, while in the case of autistic people they can mean reduced sensory and social overload [22]. Our study participants also expressed the hope that the current situation would help non-autistic people to better understand the perspective of those on the spectrum [23] and realise the consequences of what it is like to be forced to experience situations and use tools that are inadequate to the individual needs and capabilities of a given person.

## Supporting information

**S1 Table.**
(TIF)

## Acknowledgments

We thank the Participants for devoting their time, as well as their sincere and open attitude in telling own stories.

We are very grateful to Wojciech Kruszelnicki (writing and editing assistance, proofreading) and Ewa Jarosz (methodological consultation) from University of Silesia in Katowice for invaluable support in the process of writing this paper. All persons who have made substantial contributions to the work reported in the manuscript, but who do not meet the criteria for authorship, have given us their written permission to be named.

## Author Contributions

**Conceptualization:** Jacek Błeszyński, Aleksandra Rumińska, Agnieszka Hamerlińska, Renata Stefańska-Klar.

**Data curation:** Agnieszka Warszawa.

**Formal analysis:** Jacek Błeszyński, Aleksandra Rumińska, Renata Stefańska-Klar.

**Funding acquisition:** Agnieszka Warszawa.

**Methodology:** Jacek Błeszyński, Aleksandra Rumińska, Agnieszka Hamerlińska.

**Resources:** Aleksandra Rumińska, Renata Stefańska-Klar, Agnieszka Warszawa.

**Supervision:** Jacek Błeszyński, Aleksandra Rumińska, Agnieszka Hamerlińska.

**Writing – original draft:** Jacek Błeszyński, Aleksandra Rumińska.

**Writing – review & editing:** Agnieszka Hamerlińska, Renata Stefańska-Klar.

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
