## [Decision Letter · Decision Letter 0]

14 Jan 2022

PONE-D-21-24954The Experience of the COVID-19 Pandemic by Autistic People: Social AspectsPLOS ONE

Dear Dr. Błeszyński,  Please modify the manuscript as suggested by the reviewers

We look forward to receiving your revised manuscript.

Kind regards,

Soumitra Das

Academic Editor

PLOS ONE

Journal Requirements:

2. Please consider changing "autistic people” with "people with autism", including in the title (see for instance https://apastyle.apa.org/style-grammar-guidelines/bias-free-language/disability).

3. Peer review at PLOS ONE is not double-blinded (https://journals.plos.org/plosone/s/editorial-and-peer-review-process). For this reason, authors should include in the revised manuscript all the information removed for blind review.

"AUTHORSHIP STATEMENT

Manuscript title: The Experience of the COVID-19 Pandemic by Autistic People: 

Social Aspects

All persons who meet authorship criteria are listed as authors, and all authors certify 

that they have participated sufficiently in the work to take public responsibility for 

the content, including participation in the concept, design, analysis, writing, or 

revision of the manuscript. Furthermore, each author certifies that this material or 

similar material has not been and will not be submitted to or published in any other 

publication before its appearance in the Autism. 

Authorship contributions: 

Category 1 

Conception and design of study: Aleksandra Rumińska, Agnieszka Warszawa; 

acquisition of data: Aleksandra Rumińska; 

analysis and/or interpretation of data: Aleksandra Rumińska, Agnieszka 

Warszawa. 

Category 2 

Drafting the manuscript: Aleksandra Rumińska, Agnieszka Warszawa; 

revising the manuscript critically for important intellectual content: Jacek J. 

Błeszyński, Aleksandra Rumińska, Agnieszka Hamerlińska, Renata StefańskaKlar, Agnieszka Warszawa. 

Category 3 

The Research Project Manager: Jacek J. Błeszyński;

Approval of the version of the manuscript to be published (the names of all authors 

must be listed): Jacek J. Błeszyński, Aleksandra Rumińska, Agnieszka 

Hamerlińska, Renata Stefańska-Klar, Agnieszka Warszawa.

Acknowledgements 

All persons who have made substantial contributions to the work reported in the 

manuscript (e.g., technical help, writing and editing assistance, general support), 

but who do not meet the criteria for authorship, are named in the 

Acknowledgements and have given us their written permission to be named. If we 

have not included an Acknowledgements, then that indicates that we have not 

received substantial contributions from non-authors: Wojciech Kruszelnicki, Ewa 

Jarosz"

"We would also like to

thank the Committee of "CRUSH - badania nad pandemią COVID-19" contest at Nicolaus Copernicus

University in Toruń for their trust and funding our project."

"The project was implemented as part of the "CRUSH - research and COVID-19 pandemic" competition at Mikołaj

Copernicus University in Toruń"

7. Please ensure that you include a title page within your main document. You should list all authors and all affiliations as per our author instructions and clearly indicate the corresponding author.

8. Please amend either the abstract on the online submission form (via Edit Submission) or the abstract in the manuscript so that they are identical

9. Please include your full ethics statement in the ‘Methods’ section of your manuscript file. In your statement, please include the full name of the IRB or ethics committee who approved or waived your study, as well as whether or not you obtained informed written or verbal consent. If consent was waived for your study, please include this information in your statement as well. 

10. We note you have included a table to which you do not refer in the text of your manuscript. Please ensure that you refer to Table 1 in your text; if accepted, production will need this reference to link the reader to the Table.

Reviewers' comments:

Reviewer's Responses to Questions

**Comments to the Author**

1. Is the manuscript technically sound, and do the data support the conclusions?

Reviewer #1: Partly

Reviewer #2: Yes

2. Has the statistical analysis been performed appropriately and rigorously? 

Reviewer #1: N/A

Reviewer #2: Yes

3. Have the authors made all data underlying the findings in their manuscript fully available?

Reviewer #1: Yes

Reviewer #2: Yes

4. Is the manuscript presented in an intelligible fashion and written in standard English?

Reviewer #1: Yes

Reviewer #2: Yes

5. Review Comments to the Author

Reviewer #1: a. It would be better to avoid terms like “autistic people”. It sounds more stigmatizing. It would be better, if the authors use the term “persons with autism spectrum disorders”

b. Two different abstracts (one structured and one unstructured) are used by the authors. Kindly check.

c. What is the hypothesis of the study?

d. It is well known that significant proportion of people with autism spectrum disorders, have intellectual disability. Whether the authors considered the IQ of the participants before conducting interview and obtaining consent? Presence of psychopathology and severity of autism may influence the engagement in interview for assessment. This may influence the results significantly. How, these factors are taken care?

Reviewer #2: The Experience of the COVID-19 Pandemic by Autistic People: Social Aspects

--Manuscript Draft--

Manuscript Number: PONE-D-21-24954

Covid 19 pandemic has resulted in multiple psychological problems-(boredom, frustration, and embitterment), psychiatric issues (anxiety, depression, post-traumatic stress disorder, suicidal behavior and acute psychotic episodes) and social consequences (stigma, financial instability, sense of isolation etc).Studies of Covid 19 pandemic related issues have been addressed on people with Autistic Spectrum Disorder and their care givers(Yasser SaeedKhan et al,2021; Raman Baweja et al 2021, JorgeLugo-Marín et al (2021)).Several strategies have been advanced including Pfeiffer’s team in the college’s REACH Lab (Research, Engagement and Advocacy for Community Participation and Health) interventions and outcome measures aimed at promoting inclusion in community activities for those with ASD and intellectual and developmental disabilities (IDD).

This qualitative research has employed double hermeneutic of Anthony Giddens and interpretative phenomenological analysis (combination of psychological, interpretative, and idiographic components) with eight issues in ten adults diagnosed with autism spectrum, aged 25–45, out of which two as non-binary or genderqueer .Jefferson Transcription System was used in assessing participants’ statements (five recurring topics).The thematic analysis usually involves coding reliability thematic analysis, codebook thematic analysis and reflexive thematic analysis.

The case analysis reveals ‘the sense of falling out of social roles’, suicidal preoccupation, ‘craved for some interaction, but on the other hand, direct contact with people during the pandemic made her anxious’.

Digital health measures are necessary in this changing scenario as exemplified by Baweja’s article. The current article suggests ‘many autistic people seem to be well prepared for functioning in the new, rapidly emerging digital reality’. This may be true especially in high functioning autistic individuals

Additional Comments

1 Qualitative research needs small sample as in this article

2 The authors have employed probably the best methodological approach during the difficult times of covid

3 These digital methods may be incorporated into the frame work of ASD, especially in high

functioning cases

4 Further qualitative researches may help in adopting new strategies

Recommendation

1May include the recent articles, may specify high functioning ASD if necessary.

2Can be accepted

(Yasser SaeedKhan et al (2021) Research in Developmental Disabilities 119:104090;

Raman Baweja et al (2021) Journal of Autism and Developmental Disorders;

JorgeLugo-Marín et al (2021) Research in Autism Spectrum Disorders83, 101757)

6. PLOS authors have the option to publish the peer review history of their article (what does this mean?). If published, this will include your full peer review and any attached files.

Reviewer #1: No

Reviewer #2: **Yes: **Dr E.Mohandas Warrier

---

## [Author Response · Author response to Decision Letter 0]

24 Mar 2022

Manuscript Number: PONE-D-21-24954

# Reviewer 1

Dear Prof. Das,

We would like to express our sincere gratitude for instructive comments. We have incorporated most of the suggestions. Those changes are highlighted within the manuscript. Please see below, for a point-by-point response to comments. All page numbers refer to the revised manuscript file with tracked changes.

Reviewer comments to the Authors:

a. It would be better to avoid terms like “autistic people”. It sounds more stigmatizing. It would be better, if the authors use the term “persons with autism spectrum disorders”.

Authors’ response: In the manuscript, we deliberately use Identity-First Language, which is recommended by ASAN: 

https://autisticadvocacy.org/about-asan/identity-first-language/

https://autisticuk.org/wp-content/uploads/2016/05/AUTISTIC-UK-KEY-TEXTS-1-WHY-I-DISLIKE-PERSON-FIRST-LANGUAGE.pdf

https://www.ncbi.nlm.nih.gov/pmc/articles/PMC7817071/

Our team consists of autistic self-advocates researchers (Jasna Strona Spektrum, ASAN). If it is necessary to change the language, we agree, but Identity-First Language is not stigmatizing for us and we use it in our work and private life.

b. Two different abstracts (one structured and one unstructured) are used by the authors. Kindly check.

Authors’ response: Thank you for pointing this out. The reviewer is correct. Sending the abstract and lay abstract results from our mistake.

c. What is the hypothesis of the study?

Authors’ response: In qualitative social research, we do not begin the investigation from hypotheses („null hypothesis”). Research questions are formulated on the basis of the aims.

d. It is well known that significant proportion of people with autism spectrum disorders, have intellectual disability. Whether the authors considered the IQ of the participants before conducting interview and obtaining consent? Presence of psychopathology and severity of autism may influence the engagement in interview for assessment. This may influence the results significantly. How, these factors are taken care?

Authors’ response: Thank you, this is an important point. Many people on the autism spectrum suffer from anxiety disorders and depression, but we took this fact into consideration https://link.springer.com/article/10.1007/s40489-021-00267-6

We recruited participants through organizations offering diagnosis and therapy for autistic people and in order for the qualification to the sample group we conducted a self-report poll were criterion was the intellectual norm. The participants' consent was voluntary and self-conscious.

[on page 3, lines 91 to 95]

“All participants are within the intellectual norm or have supra-normative cognitive abilities. Like many people on the autism spectrum, they may have anxiety disorders or depression [13, 14, 15]. 

They were qualified on the basis of self-report, but their intellectual status was confirmed by the organizations that made contacts with each of them available.”

# Reviewer 2

Dear Dr Warrier,

Thank you for your positive review. We would like to express our sincere gratitude for instructive comments. Those changes are highlighted within the manuscript. All page numbers refer to the revised manuscript file with tracked changes. 

Reviewer comments to the Authors: 

a, b. May include the recent articles, may specify high functioning ASD if necessary. (…)

Authors’ response: Thank you for this suggestion. In line with the recommendation, we extended the references to include the recent articles: 

[on page 3, lines 91 to 93]

“All participants are within the intellectual norm or have supra-normative cognitive abilities. Like many people on the autism spectrum, they may have anxiety disorders or depression [13, 14, 15].” 

[on page 19, lines 543 to 553]

13. Baweja R., Brown S., Edwards E., Murray M. (2021), COVID-19 Pandemic and Impact on Patients with Autism Spectrum Disorder. Journal of Autism and Developmental Disorders, doi: 10.1007/s10803-021-04950-9

14. Lugo-Marín J., Gisbert-Gustemps L., Setien-Ramos I., Español-Martín G., Ibañez-Jimenez P., Forner-Puntonet M. et al (2021), COVID-19 pandemic effects in people with Autism Spectrum Disorder and their caregivers: Evaluation of social distancing and lockdown impact on mental health and general status. Research in Autism Spectrum Disorders 83:101757, doi: 10.1016/j.rasd.2021.101757

15. Khan Y.S., Khan A.W. Tahir M.E., Hammoudeh S., Shamlawi M.A., Alabdulla M. (2021), The impact of COVID-19 pandemic social restrictions on individuals with autism spectrum disorder and their caregivers in the State of Qatar: A cross-sectional study. Research in Developmental Disabilities 119:104090, doi: 10.1016/j.ridd.2021.104090

Best regards, 

Authors

---

## [Editor Report · Decision Letter 1]

4 Apr 2022

The experience of the COVID-19 pandemic by persons with ASD: Social aspects

PONE-D-21-24954R1

Dear Dr. Rumińska,

We’re pleased to inform you that your manuscript has been judged scientifically suitable for publication and will be formally accepted for publication once it meets all outstanding technical requirements.

Kind regards,

Soumitra Das

Academic Editor

PLOS ONE
---

## [Editor Report · Acceptance letter]

8 Jun 2022

PONE-D-21-24954R1 

The experience of the COVID-19 pandemic by persons with ASD: Social aspects. 

Dear Dr. Rumińska:

I'm pleased to inform you that your manuscript has been deemed suitable for publication in PLOS ONE. Congratulations! Your manuscript is now with our production department. 

Kind regards, 

on behalf of

Dr. Soumitra Das 

Academic Editor

PLOS ONE